# Recent Advances in the Research on the Anticyanobacterial Effects and Biodegradation Mechanisms of *Microcystis aeruginosa* with Microorganisms

**DOI:** 10.3390/microorganisms10061136

**Published:** 2022-05-31

**Authors:** Yun Kong, Yue Wang, Lihong Miao, Shuhong Mo, Jiake Li, Xing Zheng

**Affiliations:** 1College of Resources and Environment, Yangtze University, Wuhan 430100, China; 2021720590@yangtzeu.edu.cn; 2State Key Laboratory of Eco-Hydraulics in Northwest Arid Region, Xi’an University of Technology, Xi’an 710048, China; moshuhong@xaut.edu.cn (S.M.); xaut_ljk@163.com (J.L.); xingzheng@xaut.edu.cn (X.Z.); 3Key Laboratory of Water Pollution Control and Environmental Safety of Zhejiang Province, Hangzhou 310058, China; 4School of Biology and Pharmaceutical Engineering, Wuhan Polytechnic University, Wuhan 430023, China; miaowhpu@126.com

**Keywords:** *Microcystis aeruginosa*, microorganisms, biodegradation, anticyanobacterial modes, harmful cyanobacterial blooms

## Abstract

Harmful algal blooms (HABs) have attracted great attention around the world due to the numerous negative effects such as algal organic matters and cyanobacterial toxins in drinking water treatments. As an economic and environmentally friendly technology, microorganisms have been widely used for pollution control and remediation, especially in the inhibition/biodegradation of the toxic cyanobacterium *Microcystis aeruginosa* in eutrophic water; moreover, some certain anticyanobacterial microorganisms can degrade microcystins at the same time. Therefore, this review aims to provide information regarding the current status of *M. aeruginosa* inhibition/biodegradation microorganisms and the acute toxicities of anticyanobacterial substances secreted by microorganisms. Based on the available literature, the anticyanobacterial modes and mechanisms, as well as the in situ application of anticyanobacterial microorganisms are elucidated in this review. This review aims to enhance understanding the anticyanobacterial microorganisms and provides a rational approach towards the future applications.

## 1. Introduction

Harmful cyanobacterial blooms (HCBs) caused by cyanobacteria (including *Microcystis*, *Anabaena*, *Nodularia*, *Oscillatoria*, and so on) have become a common occurrence in freshwater worldwide [1,2]. Among the blooming cyanobacteria, *Microcystis aeruginosa* is one of the most common and widespread species [3]; specifically, it is known to be a representative species due to the dominant production of microcystins [4,5]. The rapid and excessive growth of *M. aeruginosa* is harmful to drinking water treatments and aquatic ecosystems due to the release of algal organic matters and cyanobacterial toxins [6,7]. As a result, the control of HCBs in water sources is a matter of great urgency.

Many approaches have been adopted for *M. aeruginosa* removal over the past few decades [8]. Physical methods including mechanical salvage, physical aeration, and ultrasonic treatment are usually high cost and take a long time; chemical methods such as chemical oxidants are highly efficient and low-cost methods for controlling HCBs within a short time [9]. However, chemicals may lead to a secondary contamination that may lead to potential threats to the aquatic ecosystem [10,11]. Compared with the physical and chemical methods, biological approaches such as plant allelopathy, aquatic animals and anticyanobacterial microorganisms are considered to be an economic and environmentally friendly way for cyanobacteria inhibition/biodegradation [2,10,12]. Among these methods, anticyanobacterial microorganisms are used as efficient biological agents *M. aeruginosa* [13]; furthermore, the microcystins can be biodegraded by certain anticyanobacterial microorganisms at the same time [6,14,15].

Up to now, several review articles have been published to introduce the anticyanobacterial microorganisms including bacteria, viruses, and fungi [2,10,13,16,17]. However, the previous reviews have concentrated mainly on both the freshwater and marine cyanobacterial/algal species or diatoms. While few studies have focused on elimination and degradation of the toxic cyanobacterium *M. aeruginosa* by bacteria and fungi. Moreover, the important role of anticyanobacterial microorganisms on the key genes expression and the anticyanobacterial activities regulated by quorum sensing (QS) system hasn’t been mentioned. In order to clarify the current situation of anticyanobacterial microorganisms for *M. aeruginosa* control, the available literature on the bacteria and fungi (studies that focused on bacteriophages against *Microcystis* spp. are not included in this review) are adapted to review the current progress. In this review, anticyanobacterial substances and their acute toxicities (the half maximal effective concentration, EC_50_), anticyanobacterial modes and mechanisms, as well as in situ application of anticyanobacterial microorganisms are elucidated. This review will enhance understanding the anticyanobacterial microorganisms and provide a rational attitude towards future application

## 2. Anticyanobacterial Effects for *M. aeruginosa*

### 2.1. Anticyanobacterial Microorganisms

Over the past few decades, the isolation and identification of microorganisms with anticyanobacterial effects have attracted extensive concern. Based on the literature, a variety of anticyanobacterial microorganisms have been isolated from the natural environment, and most of them belong to the anticyanobacteria and anticyanobacterial fungi.

#### 2.1.1. Anticyanobacteria

The high diversity of anticyanobacteria reported in the literatures is summarized in Table 1. There are more than 50 genera belonging mainly to Proteobacteria, Actinomycetes, Bacteroidetes, Firmicutes and Thermus. Proteobacteria, which is divided into five parts, is one of the most widespread and extensively studied bacteria in the microbiology field, and it is well known to effectively biodegrade cyanobacteria and diatoms in eutrophic environments [2,10]. The majority have been identified as members of *Pseudomonas* [18,19], *Aeromonas* [20,21], *Acinetobacter* [22], *Raoultella* [23], *Brevundimonas* [24], *Ochrobactrum* [25], *Halobacillus* [26], *Shewanella* [27], *Citrobacter* [28], *Stenotrophomonas* [29], *Serratia* [30] and *Hahella* [31] genera belonging to the γ-Proteobacteria class.

According to the microbial taxonomy, anticyanobacterial Actinomycetes can be classified into four major categories: *Streptomyces* sp. [32,33], *Rhodococcus* sp. [34], *Microbacterium* sp. [35] and *Arthrobacter* sp. [14]. *Streptomyces* is the most common anticyanobacterial Actinomycetes in HCBs control. A previous study confirmed that *S. grisovariabilis* NT0401 shows a high anticyanobacterial activity against *M. aeruginosa* by secreting active substances [36], and the anticyanobacterial substances of amino acids (L-lysine and L-valine) [3,37], tryptamine [38] and triterpenoid saponin [35] from Actinomycetes have been identified. In addition to Actinomycetes, many other Bacteroidetes are also highly efficient at inhibiting the growth of *M. aeruginosa*, such as *Aquimarina* sp. [39], *Chryseobacterium* sp. [40,41], *Aureispira* sp. [42] and *Pedobacter* sp. [43]. Although the Bacteroidetes group has been reported to inhibit cyanobacteria, diatoms and green algae [2,10], there is no publication on the inhibition of *M. aeruginosa* by *Flavobacterium* sp. or *Cellulomonas* sp.

It is shown in Table 1 that the largest number of anticyanobacterial Firmicutes are the *Bacillus* group, accounting for 77.3% of the total number of Firmicutes, while the remaining strains are from the genera *Exiguobacterium* [44,45] and *Staphylococcus* [35]. Li et al., (2015) revealed that *Bacillus* sp. Lzh-5 releases anticyanobacterial substances to attack *M. aeruginosa*, *M. viridis*, *Chroococcus* sp., and *Oscillatoria* sp. [46]; *B. licheniformis* Sp34 can also effectively destroy the cell membrane of *M. aeruginosa* and inhibit the synthesis of microcystins [47]; moreover, the simultaneous application of *Bacillus* sp. T4 and toxin-degrading bacteria could eliminate both *Microcystis* sp. and microcystins [48]. These results demonstrate that *Bacillus* not only inhibits the growth of *M. aeruginosa* [49,50], but also inhibits the expression of microcystins synthesis gene *mcyB* [47,51] and degrades the cyanobacterial toxins [48]. Obviously, *Bacillus* has a potential application for HCBs control.

There is only one strain of *Deinococcus metallilatus* MA1002 attached to Thermus that has been reported to inhibit *M. aeruginosa* [52]. The bacterium *Deinococcus* sp. also shows an anticyanobacterial effect on the toxic dinoflagellate *Alexandrium tamarense* [53]. Except for the genera mentioned above, other genera connected with anticyanobacterial or flocculation activities also exist, including *Citrobacter* sp. [28,54] and *Sphingopyxis* sp. [55]. The above anticyanobacteria can destroy the *M. aeruginosa* cells by causing cell membrane damage, and oxidative stress and by inhibiting the gene expression from a wide range of temperatures (−20 to 121 °C) and pH (3 to 11) [5,32,33]. Not only that, the photosynthesis system of *M. aeruginosa* is also reduced [56]. To summarize, the anticyanobacteria can effectively inhibit the growth of *M. aeruginosa*, and cause an inhibition effect at a low concentration.

#### 2.1.2. Anticyanobacterial Fungi

Compared with the studies of anticyanobacteria, the research and application of fungi for eliminating or inhibiting *M. aeruginosa* cells has not received much attention until 2010 [105,106]. Only Ascomycetes and Basidiomycetes have been found to have the anticyanobacterial effects against *M. aeruginosa*. It has been reported that the fungus *Trichaptum abietinum* 1302BG can eliminate four cyanobacteria directly including *M. aeruginosa* FACH-918 and *M. aeruginosa* PCC 7806 in 48 h [106]. Some other fungi such as *Trichoderma citrinoviride* [6], *Penicillium chrysogenum* [97], *Aureobasidium pullulans* KKUY070 [98], *Lopharia spadicea* [99], *Phanerochaete chrysosporium* [100,101], *Irpex lacteus* T2b [102], *Trametes versicolor* F21a [107] and *Bjerkandera adusta* T1 [103] also show good inhibitory activities against *M. aeruginosa*. It has been stated that *T. citrinoviride* and *A. pullulans* have highly specific anticyanobacterial effects towards *Microcystis* spp. while they have an insignificant effects on the green algae or diatoms [6,98]; furthermore, the biodegradation of *M. aeruginosa* cells may be due to the excretion of the lytic enzyme (N-β-acetylglucosaminidas) [98], which can degrade the peptidoglycan from the cyanobacterial cell wall. The extracellular enzymes of cellulase, β-glucosidase, protease, and laccase from *T. versicolor* F21a have also been proven to be responsible for the degradation of *Microcystis* spp. [107,108].

On the contrary, the *M. aeruginosa* cells are damaged in a short time under the treatment of *T. abietinum* 1302BG, *I. lacteus* T2b or *T. hirsuta* T24, and the anticyanobacterial process occurs “cell to cell” through the following steps: (1) the fungus comes into physical contact with the surface of the cyanobacterial cells; (2) cyanobacterial cells are encompassed with mycelia, which destroy the cyanobacterial cell wall and membrane; and (3) the nucleic acids and other substances of cyanobacteria cells are released [17]. Fungi have the natural ability to destroy *Microcystis* cells by secreting anticyanobacterial substances or through “cell to cell” contact. Apart from the growth inhibition and cell lysis of *M. aeruginosa*, some fungi are able to remove microcystins [6,98,106], and the removal mechanism is related to the adsorption/biodegradation of fungus or the inhibition expression of microcystins synthesis gene [15].

### 2.2. Anticyanobacterial Substances

The metabolic activities of microorganisms are diverse, some of the secretory substances have anticyanobacterial or algicidal activities. However, due to the complexity of separation and purification, only part of the anticyanobacterial substances have been identified [2,10]. On the basis of the relative literatures and types of compounds, the isolated substances can be classified into five major categories: alkaloids, protein/amino acids, fatty acid/cyclic peptides/peptide derivates, enzymes and others (Table 2). The alkaloids are not only secreted by bacteria such as *Aeromonas* sp. [67,69], *Pseudomonas* sp. [66], *Bacillus* sp. [88,91] and *Streptomyces* sp. [38,84], but are also produced by the fungus *Phellinus* sp. [104]. For example, the anticyanobacterial compound isolated from *A. guillouiae* A2 has been identified as 4-hydroxyphenethylamine (C_8_H_11_NO), with the EC_50,72h_ of 22.5 ± 1.9 mg L^−1^ in 72 h [72]; the prodigiosin can be produced by both *S. marcescens* LTH-2 and *Hahella* sp. KA22, while it shows higher anticyanobacterial effect against *M. aeruginosa* FACHB 905 (EC_50,72h_ of 0.16 mg L^−1^) compared to *M. aeruginosa* FACHB-1752 (EC_50,72h_ of 5.87 mg L^−1^) [31,109], demonstrating the different EC_50_ of prodigiosin is probably related to the cyanobacteria species. For the cyclic peptides, the hexahydropyrrolo[1,2-a]pyrazine-1,4-dione (cyclo[Gly-Pro]) can also be secreted by *Stenotrophomonas* sp. [29], *Bacillus* sp. [46] and *Shewanella* sp. [27], the EC_50,24h_ against *M. aeruginosa* 9110 is from 5.7 to 5.9 mg L^−1^.

The diketopiperazine substances produced by bacteria have been recognized as having anticyanobacterial activities for *M. aeruginosa*. The EC_50,24h_ value of cyclo(4-OH-Pro-Leu) (7-hydroxy-3-isobutyl-hexahydro-pyrrolo[1,2-a]pyrazine-1,4-dione) and cyclo(Pro-Leu) (hexahydro-3-(2-methylpropyl)-pyrrolo[1,2-a]pyrazine-1,4-dione) isolated from *Chryseobacterium* sp. GLY-1106 against *M. aeruginosa* is 1.26 and 2.70 mg L^−1^, respectively [41]. Another diketopiperazine 3-benzyl-piperazine-2,5-dione (cyclo[Gly-Phe]) was firstly reported by Guo et al., (2016) [69], who showed that cyclo(Gly-Phe) has weaker anticyanobacterial activity (EC_50,24h_ of 4.72 mg L^−1^) compared with cyclo(Pro-Phe) (EC_50,24h_ of 1.85 mg L^−1^) [88]. Diketopiperazine substances with similar structures often exhibit distinct biological properties. After short-term exposure to *M. aeruginosa*, cyclo(4-OH-Pro-Leu) interrupts the flux of electron transport in the photosynthetic system and cyclo(Pro-Leu) inhibits the antioxidant enzyme activities of *M. aeruginosa* [41], whereas 3-isopropyl-hexahydropyrrolo[1,2-a]pyrazine-1,4-dione (cyclo[Pro-Val]) causes significant damage to cyanobacterial cell membranes [46].

Previous studies have indicated that amino acids have powerful anticyanobacterial effects against *Microcystis* spp. at concentrations between 0.6 and 5.0 mg L^−1^ [11,110,111], and the inhibition effect of L-lysine against *Microcystis* sp. is remarkable [110]. Moreover, the eutrophic lake with the dominant species of cyanobacterium *M. aeruginosa* is selectively controlled by lysine [111]. The amino acids and proteins have commonly been identified and reported as the anticyanobacterial substances for *M. aeruginosa*. Two amino acids (L-lysine and L-phenylalanine) are purified from *B. amyloliquefaciens* T1 that have an inhibition effect against *M. aeruginosa* FACHB-905 [94]; the L-valine, which shows a better anticyanobacterial activity than L-lysine, is also isolated from *S. jiujiangensis* JXJ 0074 [37]. It is interesting that the anticyanobacterial efficiency of tryptamine and tryptoline on *M. aeruginosa* FACHB-905 is 80 ± 1% and 100 ± 2%, respectively, but the growth of *M. aeruginosa* is recovered as tryptamine (tryptoline) and is completely used or degraded by microorganisms [38]. Therefore, the persistence of amino acids should be further considered when they are used for eutrophication control [112].

## 3. Anticyanobacterial Modes and Mechanisms

### 3.1. Anticyanobacterial Modes

In general, the anticyanobacterial modes by microorganisms are divided into direct attack (bacterial and cyanobacterial cell contact) and indirect attack (the release of anticyanobacterial substances) (Figure 1) [10,32,72,118]. To date, although anticyanobacteria can directly kill several different kinds of cyanobacteria, only few has been reported. A wide range of cyanobacteria including *M. aeruginosa*, *M. wesenbergii*, *M. viridis*, *Anabaena flos-aquae*, *Oscillatoria tenuis*, *Nostoc punctiforme* and *Spirulina maxima* are lysed by *B. cereus* DC22 with the direct attack mode, as well as chlorophyceae (*Chlorella ellipsoidea* and *Selenastrum capricornutum*) [89]. In addition to *B. cereus*, other anticyanobacteria that destroy *M. aeruginosa* with direct attack have also been reported. For example, the anticyanobacterial modes of *Aeromonas bestiarum* HYD0802-MK36 [20], *Chryseobacterium* sp. [40], *Streptomyces globisporus* G9 [83], *Alcaligenes denitrificans* [59], and *Shigella* sp. H3 [60] on *M. aeruginosa* are regarded as direct attack, and a number of cyst-like cells are formed in cyanobacteria during the direct attack [10]. It is speculated that the cyanobacterial cell walls are partially destroyed at the contact point with the anticyanobacteria, and the formation of cyst-like cells is a potential defense system against anticyanobacteria [2,10].

The indirect attack mode has been observed in the numerous metabolites from most of the reported anticyanobacterial microorganisms, and the anticyanobacterial characteristics of these bacteria seem to be unique to *M. aeruginosa*. Up to now, the genus *Acinetobacter* [22,72,119] and *Exiguobacterium* [44,45,96], which firstly attach to *M. aeruginosa* and then cause serious damage to the cyanobacterial cell structure and morphology, are recognized as degrading *M. aeruginosa* by producing anticyanobacterial substances. Nevertheless, some anticyanobacteria can inhibit or kill green alga and cyanobacteria with an indirect attack simultaneously. For instance, *B. amyloliquefaciens* FZB42 can efficiently eliminate *M. aeruginosa*, *Anabaena* sp., *A. flos-aquae* and *Nostoc* sp. by secreting bacilysin [91]. In line with this genus, *B. amyloliquefaciens* T1 produces amino acids to inhibit the growth of four *Microcystis* spp., but not of *Anabaena flos-aquae* or *Chlorella pyrenoidosa* [49,94]; *S. amritsarensis* HG-16 kills *A. flos-aquae*, *Phormidium* sp. and five *Microcystis* spp. by secreting active substances, but has a small inhibitory effect on *C. vulgaris* and a promoting effect on *Oscillatoria* sp. [5]. Along with this, the anticyanobacterial modes of *Aquimarina salinaria* on green algae and cyanobacterium, which is a direct attack on *C. vulgaris* 211-31 and an indirect attack on *M. aeruginosa* MTY01, is quite different [39]. Furthermore, a recent study firstly demonstrated that *Paucibacter aquatile* DH15 inhibits *M. aeruginosa* by both direct and indirect attacks [61], which would be interesting and could shed further light on the anticyanobacterial modes by microorganisms.

### 3.2. Anticyanobacterial Mechanisms

Currently, the anticyanobacterial mechanisms of microorganisms against *M. aeruginosa* are mainly dependeent on the attack modes, and these mechanisms are revealed with the changes in the photosynthesis system, antioxidant enzymes system, gene expression and QS system (Figure 2).

#### 3.2.1. Effects of Anticyanobacterial Microorganisms on Photosynthesis

Photosynthesis, which converts solar energy into chemical energy through the photosynthesis system (PS) II and PS I, is the principal mode of energy metabolism in cyanobacteria [120]. Anticyanobacterial microorganisms can significantly affect the photosynthesis of *M. aeruginosa* cells in several ways, including decreasing the chlorophyll *a* (Chl *a*) contents and photosynthetic pigments [56], and the disruption of the electron transport pathway in PS [23,93]. Chl *a* is one of the important components of cyanobacterial pigments. It is markedly decreased in *M. aeruginosa* under the exposure of anticyanobacteria such as *P**. aeruginosa* [18,63], *Streptomyces* sp. [33,36], *Exiguobacterium* sp. [44,45], and so on. For the photosynthetic pigments, phycocyanobilin (PC), allophycocyanin (APC) and phycoerythrin (PE) are major indicators of cyanobacterial photosynthetic efficiency and are essential apparatus for light harvesting [61], and the addition of anticyanobacterium results in a significant decrease in the PC, APC and PE by disrupting the synthesis of an photosynthetic pigments [56]. In addition, the expressions of *pcA* and *apcA* genes for PC and APC synthesis in *M. aeruginosa* are down-regulated by *Paucibacter aquatile* DH15, which shows an inhibition effect on active chlorophyll [61]. It has been noted that the Chl *a* decrease is closely related to the reduction in photosynthetic pigments, and the cyanobacterial membrane is sensitive and easily damaged by anticyanobacterium [56].

The variations of cyanobacterial energy kinetics have also been evaluated by Chl fluorescence parameters, such as the maximum photochemical quantum yield of PS II (Fv/Fm), the effective quantum yield (Φe), and the maximum electron transport rate (ETRmax) [41,95]. With the addition of fermentation filtrate (5%, *v*/*v*) of *Paenibacillus* sp. SJ-73, the Fv/Fm values of *M. aeruginosa* PCC7806 and *M. aeruginosa* TH1701 dramatically decline from 0.52 and 0.29 to 0 [95]; similarly, it is only 0.08 (14.3% of the initial value) for *M. aeruginosa* FACHB-905 after being treated for 24 h by the fermentation filtrate (5%, *v*/*v*) of *Raoultella* sp. S1 [23]. Besides, the Φe and ETRmax of *M. aeruginosa* 9110 following the treatment of *Chryseobacterium* sp. GLY-1106 decrease gradually with time [41]; the ETRmax values of *M. aeruginosa* are also depressed significantly under the stress of *Raoultella* sp. S1 [23] and *Bacillus* sp. B50 [93]. The decreases in Fv/Fm, Φe and ETRmax demonstrate that the photosynthetic system is seriously damaged and the electron transport chain is blocked, resulting in the inhibition of cyanobacterial cell photosynthesis [55]. In consequence, the possible mechanism underlying the photosynthetic reduction could be due to the reduction in Fv/Fm, Φe and ETRmax in *M. aeruginosa*.

#### 3.2.2. Effects of Anticyanobacterial Microorganisms on Antioxidant Enzymes System

The oxidative damage of the cyanobacterial cells can occur under different environmental stress conditions, and it will results in an increase in reactive oxygen species (ROS), which includes the superoxide anion radical, hydrogen peroxide and hydroxyl radicals [51,61]; while excess ROS often leads to oxidative stress, lipid peroxidation, and DNA damage [56,121]. The enzymatic antioxidants (such as catalase (CAT), superoxide dismutase (SOD), peroxidase (POD), and so on) and non-enzymatic antioxidants (such as ascorbic acid (AsA) and glutathione (GSH)) are responsible for removing the overproduction of ROS [2,31,41]. For instance, *Streptomyces eurocidicus* JXJ-0089 inhibits the growth of cyanobacterial cells in various ways, including promoting ROS production (e.g., O_2_•^−^), inhibiting the antioxidant synthesis, removing chlorophyll and destroying cell walls [38].

The ROS of cyanobacteria increases excessively by either the direct attack or indirect attack of anticyanobacterial microorganisms. The O_2_•^−^ content in *M. aeruginosa* cells is induced largely by 4 μg mL^−1^ 3, 4-dihydroxybenzalacetone (DBL) secreted from *Phellinus noxius* HN-1 and increased from 0.360 ± 0.001 to 0.400 ± 0.001 μg g^−3^ [104]. The ROS level of *M. aeruginosa* NIES 843 treated with *Bacillus* sp. AF-1 (cell-free filtrate) was lower than that of the control at the first 48 h but much higher at 72 h, indicating that some evasive mechanisms were taken to prevent the ROS accumulation in cyanobacterial cells at the initial stage [51]. Similar variations of ROS have been observed in *M. aeruginosa* KW after being treated with *Paucibacter aquatile* DH15, and the malondialdehyde (MDA) content and SOD activity related to remove ROS also increased at first and then decreased [61]; The MDA content, CAT and POD activity of *M. aeruginosa* FACHB-905 also increased quickly when fermentation liquid (5%, *v*/*v*) of *P. aeruginosa* [18] and *P. chrysosporium* was added quickly [101]; moreover, the responses of *M. aeruginosa* FACHB-905 cells to *Streptomyces* sp. KY-34 and *Streptomyces* sp. HJC-D1 following a similar pattern with the increases of CAT, SOD and POD, and the MDA further increased during the incubation time [56,121]. Although the antioxidants increased immediately to relieve the damage caused by anticyanobacteria, the cyanobacterial cell membrane may have decompose due to the accumulation of MDA [18,67,121].

For the non-enzymatic antioxidants, the variation of GSH is opposite to that of the antioxidase activity. The *Bacillus licheniformis* Sp34 induces more GSH production in *M. aeruginosa* at first to clear ROS, but the GSH content is much lower at 20 h (compared with the control) [47]. Such a phenomenon is also obtained in the anticyanobacterial process of *Raoultella* sp. S1 [23]. The prodigiosin from *Hahella* sp. KA22 also leads to the variation of GSH content, while the GSH content decreases slightly after exposure for 36 h [31]. These results demonstrate that the ROS levels and MDA contents decrease under prolonged exposure to anticyanobacteria [31,33,65]; in addition, the non-enzymatic antioxidants also play a critical role in protecting the cyanobacterial cells from oxidative damage under anticyanobacterial stress [23].

#### 3.2.3. Effects of Anticyanobacterial Microorganisms on Gene Expression

The relative transcriptional level of some critical genes in cyanobacteria can be dramatically changed by anticyanobacterial microorganisms and substances, including genes related to the synthesis of photosystem reaction center proteins (*PsaA*, *psaB*, *psbA1* and *psbD1*) [47,57], peptidoglycan synthesis (*glmS*), membrane proteins (*ftsH*), antioxidase (*prx*) [100], heat-shock proteins (*grpE*) [100], fatty acids (*fabZ*) [100], cyanotoxin microcystins (*mcyA*, m*cyB*, m*cyC* and m*cyD*) [83,97], the functions of cell division (*ftsZ*) [93], CO_2_ fixation (*rbcL*) [61], and DNA repair (*ftsH* and *recA*) [2,5]. Researchers have reported that the transcription expressions of genes *ftsZ*, *psbA1*, and *glmS* are decreased by DBL that is isolated from *P. noxius* HN-1 [104] and bacilysin that secreted from *B. amyloliquefaciens* FZB42 [91]. The expressions of gene *ftsZ* and *psbA* are also significantly inhibited by *Bacillus* sp. B50 [93], and the transcriptions of photosynthesis-related genes (*psaB* and *psbD1*) and CO_2_ fixation gene (*rbcL*) are inhibited by *B. licheniformis* Sp34 [47], indicating that the metabolisms of *M. aeruginosa* are destroyed. Other studies on transcriptomic analysis have demonstrated that the principal subunits of the reaction center (*PsaA* and *PsaB*) and other subunits (*PsaC*, *PsaE*, *PsaD*, *PsaF* and *PsaL*) are significantly down-regulated by *B. laterosporus* Bl-zj [57]. It is similar in the case of *S. globisporus* G9, *S. amritsarensis* and *Raoultella* sp. S1, which suppresses the expression of *psbA1, psbD1* or *rbcL* [5,23,83]. The reduction in photosynthesis-related gene transcripts might result in an interruption in the electron transport chain and may finally affect the CO_2_ fixation process [61].

Gene such as *mcyB* that are involved in microcystins synthesis are also inhibited by *Penicillium* spp. [97], the white-rot fungi *P. chrysosporium* [100,101] and *P. noxius* HN-1 [104]; moreover, both directly attack the anticyanobacterium (*S. globisporus* G9) [83] and indirectly attack anticyanobacteria (including *S. amritsarensis* HG-16 and *Bacillus* sp. AF-1) could inhibit microcystins synthesis [5,51]. However, the inhibiting ability of *Bacillus* sp. AF-1 has not been confirmed with microcystins measurements [5].

#### 3.2.4. Regulating the Anticyanobacterial Activity by QS System

QS system is the regulator control system for microorganisms that sense the cell density of their own species and make themselves to coordinate gene expression and physiological accommodation on a community scale [122,123]. It is a cell-to-cell communication that relies on the signal molecules [124], and the accumulated QS signals can bind to the cognate receptors and regulate biological activities and cellular functions [69,125]. Previous studies have shown that microbial behaviors such as the secondary metabolites, cell motility and antibiotic resistance are all influenced by QS [122,123]; in addition, QS signals that contribute to the interactions between planktonic microalgae and bacteria are summarized as the N-acyl-homoserine lactones (AHLs) [69], the 2-alkyl-4-quinolones (AQs) [123], long-chain fatty acids and fatty acid methyl esters (autoinducer-2, AI-2) and dihydroxypentanedione furanone derivates [12]. It is agreed that most of the anticyanobacterial activities by Gram-negative bacteria (such as *Pseudomonas* sp., *Acinetobacter* sp., etc.) are the consequence of bacterial-cyanobacterial QS rather than bacterium-cyanobacteria interactions [12,124]. Some species of *Serratia* sp. [109] and *Hahella* sp. [31] can produce prodigiosin to inhibit *M. aeruginosa*, and the prodigiosin production is regulated by *LuxI* and *LuxR*, which are the crucial genes of AHLs [126]. The QS signal molecule (C4-HSL), which belongs to the classic AHL-based *LuxIR*-type QS system of Gram-negative bacteria, is responsible for the synthetic process of the anticyanobacterial compound (3-methylindole) from *Aeromonas* sp. GLY-2107 [69]. During the anticyanobacterial process, the QS systems of Gram-negative bacteria produce AHLs signaling molecules, which are synthesized by the basic regulatory protein of *LuxI* [69,88,126].

In contrast, a wide range of the Gram-positive anticyanobacteria (such as *Streptomyces* sp., *Bacillus* sp., etc.) generally use AI-2 as the signal molecules in QS systems [125]. The anticyanobacterium *S. xiamenensis* Lzh-2 exhibits QS behavior, and the *LuxS* gene is crucial for the AI-2 type QS system; obviously, the anticyanobacterial activity of *S. xiamenensis* Lzh-2 is regulated through the *LuxS*/AI-2 QS system by inducing the production of anticyanobacterial compounds 2, 3-indolinedione and cyclo(Gly-Pro) [126]. The AI-2 type QS behavior is present in *Bacillus* sp. [127]. Genomic analysis of *B. subtilis* JA has indicated the existence of the *LuxS* gene that regulates the pheromone biosynthesis, and the high-molecular-weight anticyanobacterial compounds (>3 kDa) produced by *Bacillus* sp. S51107 have been proven to be primarily regulated by the *NprR*-*NprX*-type (AI-2) QS system [88]. As a consequence, the AI-2 QS system has been considered as a possible strategy to regulate the behavior of the anticyanobacterial effects of Gram-positive bacteria. Although QS behavior has been reported in recent years, there is still an improved understanding of the interaction between cyanobacteria and anticyanobacterial microorganisms.

## 4. Application and Prospective

### 4.1. Application of Anticyanobacterial Microorganisms

In consideration of the drawbacks of physical and chemical methods, the biological control of HCBs is of great importance for the aquatic ecological environment. In particular, the application of anticyanobacterial microorganisms (bacteria and fungi) or their anticyanobacterial substances is regarded as the most suitable approach due to the economical and environment-friendly performance. It is well known that it is difficult for microorganisms to exist persistently in the aquatic environment [128]. To overcome this limitation, microbial immobilized technology using different porous matrices for enhancing the cyanobacterial removal efficiency has been attempted. For example, a biological treatment system equipped with coconut packing carriers has been established to enrich anticyanobacteria. The results indicate that the average anticyanobacterial efficiency of 87.69 ± 2.44% is obtained and 13 genera anticyanobacteria, which account for 10.17% of the total bacteria, are responsible for the removal of HCBs [129]. As the *Brevundimonas* sp. AA06 is immobilized using polyvinyl alcohol-sodium alginate beads and *B. methylotrophicus* ZJU is immobilized with Fe_3_O4 nanoparticles, the inhibition effects are much better than freely suspended cells [24,50]; meanwhile, the extracellular polymeric substances produced by *P. aeruginosa* ZJU1 are made as bioflocculants, and the removal efficiency of *M. aeruginosa* reached 100 ± 0.07% in 5 min at the dosage of 2.75 g/L bioflocculant [130]. These strategies demonstrating the “indirect attack” of microorganisms could be immobilized by multi-functional systems and their anticyanobacterial products could be further enriched. Taking full account of the uncertainties of using anticyanobacterial microorganisms to control/eliminate HCBs in natural waters, the “direct attack” microorganisms may be as ineffective as “indirect attack” microorganisms in actual applications.

In situ eutrophication controls have also been carried out in other researche. It was found that the Chl *a* removal efficiency reached 99.2% when the anticyanobacterium *B. cereus* N-1 was immobilized with a floating carrier for natural eutrophication water [48]; the wild cyanobacteria from a shallow eutrophic pond were significantly controlled by adding solid *B. amyloliquefaciens* T1 agent at the concentration of 0.5 mg L^−1^ (or above) [49]. Taking the recycling utilization of the industrial waste product into account, approximately 80.0% of the *M. aeruginosa* and 48.1% of the microcystin-LR were removed by the biosorbent, which originated from the *Escherichia coli* biomass [131]. Apart from the persistent existence of microorganisms, anticyanobacterial effects are concerned with environmental conditions and nutrient concentrations [132]. As the previous study indicates, the yeast *Candida utilis* F87, which converts the nitrogen and phosphorus into microbial protein, can inhibit the growth of *M. aeruginosa* by nutrient competition [133]. Therefore, the issue of nutrient competition in cyanobacterial control using microorganisms is a crucial consideration. Based on the current collection of literature, the anticyanobacterial microorganisms have a potential application for HCBs control in the natural environment.

### 4.2. Summary and Prospective

Interactions between cyanobacteria and microorganisms are considered to be an integral part of the geochemical cycle. However, with the spatial and temporal heterogeneity, these interactions can be modulated in various ways, and highly efficient anticyanobacterial strategies in the eutrophic environment can be obtained from microorganisms. Plentiful studies have reported on ecological interactions between anticyanobacteria and cyanobacterium *M. aeruginosa*, which are focused on the anticyanobacterial microorganisms, substances, modes and mechanisms. Although the anticyanobacterial approach by microorganisms seems to be safe and effective, it is still appreciated that there are limitations and challenges in field applications. A drawback of this approach is that anticyanobacterial microorganisms must be chosen carefully to secrete specific anticyanobacterial compounds and the dosage of the microorganism inoculum or microbial agent is of great importance. On the other hand, the abiotic and biotic factors of the natural environment may have a remarkable influence on the distribution of cyanobacteria and the cyanobacterial response to anticyanobacterial substances.

Besides the target specificity, the complicating factors in realistic eutrophic environment research are the complexity of consortia with multiple species and the unsustainability of anticyanobacteria. It is delightful to see that the studies for HCBs control in situ have contributed to a better understanding of the role of anticyanobacterial microorganisms, especially the multiple regulations for microcystins. Further investigations should be focused on the simultaneous removal of nitrogen, phosphorus and microcystins by mixed microbial community, and the understanding of the cell-to-cell communication and the defense mechanisms of QS systems. Besides, more insights are needed for the specific genes encoding photosystem synthesis, peptidoglycan synthesis, membrane proteins, cyanotoxin microcystins, DNA repair and so on.

## Figures and Tables

**Figure 1 microorganisms-10-01136-f001:**
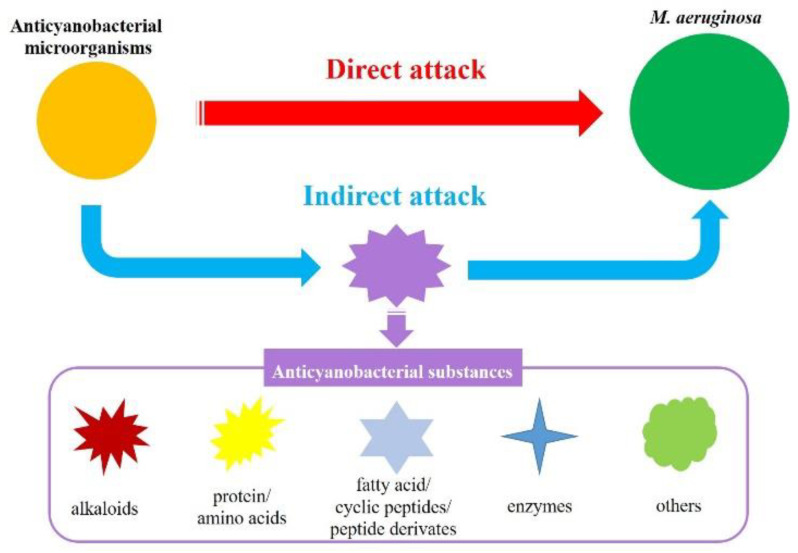
Anticyanobacterial modes of microorganisms against *M. aeruginosa*.

**Figure 2 microorganisms-10-01136-f002:**
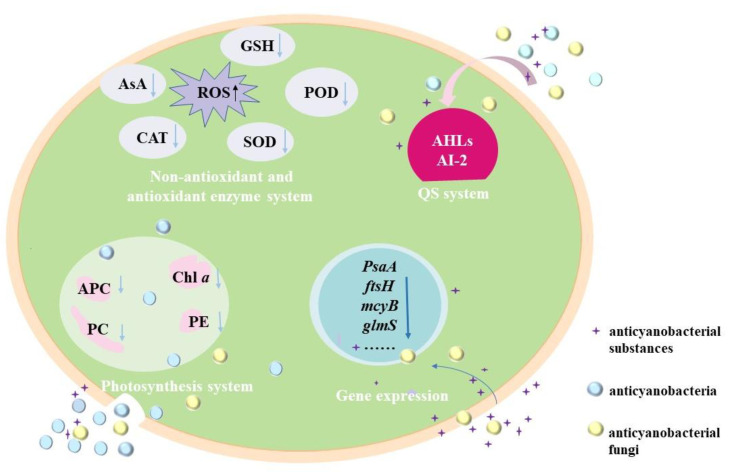
Anticyanobacterial mechanisms of microorganisms against *M. aeruginosa*.

**Table 1 microorganisms-10-01136-t001:** Summary of anticyanobacterial microorganisms and their anticyanobacterial modes.

	Strain Name	Target Cyanobacterium	Initial Cyanobacterial Cell Density (cells mL^−1^)	Dosage (*v*/*v*)	Duration Time	Inhibition Rate/Removal Efficiency	Anticyanobacterial Modes	References
**α-Proteobacteria**	*Brevibacillus laterosporus* Bl-zj	*M. aeruginosa* FACHB- 905	1.0 × 10^7^	1.0 × 10^7^ **	3 d (4 d)	72.36% (92.30%)	NA	[57]
	*Brevundimonas* sp. AA06	*M. aeruginosa* FACHB-905	2.0 × 10^9^	NA	4 d	70%	NA	[24]
	*Ochrobactrum* sp. FDT5	*M. aeruginosa*	2.0~6.0 × 10^6^	4.0 × 10^7^ **	5 d	58.9%	indirect attack	[25]
	*Stappia* sp. F2	*M. aeruginosa* FACHB-905	2.5 × 10^6^	10%	7 d	94.9%	indirect attack	[35]
	*Rhizobium* sp. AQ_MP	*M. aeruginosa*	NA	9%	10 d	100%	NA	[58]
**β-Proteobacteria**	*Alcaligenes denitrificans*	*M. aeruginosa* NIES 298	2 × 10^5^	0.7%	4 d	96.4%	direct contact	[59]
	*Alcaligenes* sp. H3	wild cyanobacterium	NA	20%	4 d	93%	indirect attack	[60]
	*Paucibacter aquatile* DH15	NA	1.0 × 10^6^	NA	36h	94.9%	combination of direct and indirect attacks	[61]
	*Achromobacter* spp. LG1	*M. aeruginosa* CAAT 2005-3	1.0 × 10^5^	1.0 × 10^6^ **	7 d	29.0 ± 1.8~55.0 ± 3.8%	NA	[62]
		*M. aeruginosa* 24A				25.3 ± 2.2~ 48.3 ± 5.5%		
**γ-Proteobacteria**	*Pseudomonas aeruginosa*	*M. aeruginosa* FACHB-905	437 ± 21 *	5% (10%)	7 d	81.21% (83.84%)	NA	[18]
	*P. aeruginosa* ACB3	*M. aeruginosa* FACHB-912	0.55~1.13 × 10^6^	1.0 × 10^7^ **	6 d	96.5%	NA	[63]
		*M. aeruginosa* FACHB-924				82.6%		
	*P. aeruginosa* UCBPP-PA14	*M. aeruginosa* NIES 298	1.0 × 10^5^	1.0 × 10^5^ **	10 d	82.4 ± 2.4%	NA	[64]
		*M. aeruginosa* NIES 44				75.0 ± 2.7%		
		*M. aeruginosa* NIER 101				69.0 ± 3.7%		
		*M. aeruginosa* NIER 100001				67.0 ± 4.2%		
	*P. grimontii* A01	*M. aeruginosa* FACHB-905	1.0 × 10^7^	10%	7 d	91.81%	NA	[65]
	*P. grimontii* A14					78.25%	NA	
	*P. putida* CH-22	*M. aeruginosa* FACHB-905	5.3 × 10^6^	15%	7 d	98.8%	indirect attack	[19]
	*Pseudomonas* sp. K44-1	*M. aeruginosa* NIES 299	NA	NA	NA	NA	indirect attack	[66]
	*P. syringae* KACC10292^T^	*M. aeruginosa* NIES 298	1.1 × 10^5^	10%	10 d	96%	indirect attack	[20]
	*Aeromonas bestiarum* HYD0802-MK36					91%	direct attack	
	*Aeromonas* sp. FM	*M. aeruginosa*	NA	5% (10%)	9 d	70.7% (88.1%)	indirect attack	[67]
	*Aeromonas* sp. FM	*M. aeruginosa* FACHB 927	1.4 × 10^7^	2.1 × 10^9^ **	4 d	up to 85%	NA	[68]
		*M. aeruginosa* FACHB 975	5.88 × 10^6^		7 d	91.2%	NA	
	*Aeromonas* sp. GLY-2107	*M. aeruginosa* 9110	1.0 × 10^7^	1%	6 d	96.5±1.1%	indirect attack	[69]
		*M. aeruginosa* PCC 7806				88.9±1.9%		
	*Aeromonas* sp. L23	*M. aeruginosa* UTEX LB 2385	6.0 × 10^6^	25%	5 d	88 ± 1.2%	indirect attack	[21]
		*M. aeruginosa* NHSB				94 ± 2.6%		
	*Aeromonas* sp.	NA	NA	8%	5 d	95%	indirect attack	[70]
	*Acinetobacter* sp. J25	NA	NA	10%	24 d	87.86%	NA	[71]
	*Acinetobacter* sp. CMDB-2	*M. aeruginosa* FACHB-905	1.0 × 10^6^	5%	3 d	87.5%	indirect attack	[22]
	*A. guillouiae* A2	*M. aeruginosa* FACHB-905	~1.0 × 10^6^	10%	7 d	91.6%	indirect attack	[72]
	*Raoultella* sp. R11	*M. aeruginosa* FACHB-905	NA	15% (30%)	6 d	57.63% (93.58%)	NA	[73]
	*R. planticola*	*M. aeruginosa* FACHB-905	NA	4% (8%)	9 d (3 d)	nearly 60% (83%)	indirect attack	[70]
	*R. ornithinolytica* S1	*M. aeruginosa* FACHB-905	NA	5%	3 d	96.2%	indirect attack	[23]
	*Halobacillus* sp. H9	*M. aeruginosa* PCC 7806	2.0 × 10^7^	5%	24h	90% (93 ± 1%)	indirect attack	[26]
		*M. aeruginosa* TAIHU98				87 ± 2%		
	*Shewanella* sp. Lzh-2	*M. aeruginosa* 9110	1.0 × 10^7^	10%	6 d	92.3 ± 6.8%	indirect attack	[27]
		*M. aeruginosa* PCC 7806				84.9 ± 3.8%		
	*Stenotrophomonas maltophilia* 15	*M. aeruginosa* FACHB-905	400 *	NA	16 d	~80%	indirect attack	[74]
	*Hahella* sp. KA22	*M. aeruginosa* FACHB-1752	NA	0.01 ***	3 d	60%	indirect attack	[31]
	*Citrobacter* sp. R1	*M. aeruginosa* FACHB-905	1.0 × 10^7^	16.7%	3 d	81.6 ± 2.2%	NA	[28]
	*Citrobacter* sp. AzoR-1	*M. aeruginosa*	1.0 × 10^7^	NA	NA	~95%	indirect attack	[54]
	*Enterobacter* sp. NP23	*M. aeruginosa*	1.0 × 10^8^	1.0 × 10^8^ **	20 d	~70%	NA	[75]
	*Shigella* sp. H3	wild cyanobacterium	NA	20%	10 d	76%	direct attack	[60]
	*Serratia marcescen*s LTH-2	*M. aeruginosa* TH1	3.0 × 10^6^	5%	2 d (3 d)	72.4% (79.0%)	indirect attack	[76]
		*M. aeruginosa* TH1				70.0% (74.6%)		
		*M. aeruginosa* FACHB-905				84.3% (87.7%)		
	*S. marcescens* BWL1001	*M. aeruginosa*	NA	NA	2 d	91.1%	indirect attack	[30]
	*Aquimarina salinaria*	*M. aeruginosa* MTY01	1.0 × 10^5^	10%	3 d (6 d)	80% (100%)	indirect attack	[39,77]
	*Chryseobacterium* sp.	*M. aeruginosa* FACHB-905	6.0 × 10^6^	10%	3 d	up to 80%	direct attack	[40]
**Bacteroidetes**	*Chryseobacterium* sp. H2	*M. aeruginosa* FACHB-905	NA	10%	7 d	85.3%	NA	[78]
	*Chryseobacterium* sp. GLY-1106	*M. aeruginosa* 9110	1.0 × 10^7^	NA	6 d	98.9%	indirect attack	[41]
	*Chryseobacterium* sp. S7	*M. aeruginosa* FACHB-905	718 *	28.5%	7 d	59.37%	indirect attack	[79]
	*Aureispira* sp. CCB-QB1	*M. aeruginosa* NISE 102	NA	NA	3min	75.39%	indirect attack	[42]
	*Pedobacter* sp. Mal 11-5	*M. aeruginosa* NIES 843	NA	6.7%	2 d (10 d)	exceeded 50% (75~85%)	NA	[43]
**Actinomycetes**	*Streptomyces* sp. NT0401	*M. aeruginosa* PCC 7806	NA	5%	5 d	up to 85%	indirect attack	[36]
		*M. aeruginosa* XW01						
	*Streptomyces* sp. L74	*M. aeruginosa* FACHB-905	1.0 × 10^6^	10%	4 d	71.48 ± 5.33%	indirect attack	[33]
	*S. neyagawaensis*	*M. aeruginosa* NIES 298	NA	NA	7 d	84.5%	NA	[80]
	*S. rameus* KKU-A3	*M. aeruginosa* KKU-13	NA	10%	7 d	81.56%	NA	[81]
	*S. aurantiogriseus* PK1	*M. aeruginosa* KKU-13	~1.5 × 10^6^	5%	8 d	~83.3%	indirect attack	[82]
	*Streptomyces* sp. KY-34	*M. aeruginosa* FACHB-905	354.3 ± 13.8 *	3% (10%)	8 d	81.2% (99.0%)	indirect attack	[56]
	*Streptomyces* sp. HJC-D1	*M. aeruginosa* FACHB-905	637.5 ± 32.1 *	5% (10%)	5 d	88.4 ± 2.8% (91.8 ± 1.2%)	indirect attack	[32]
	*S. globisporus* G9	*M. aeruginosa* NIES 44	300 ± 60 *	5%	5 d	95.1 ± 1.6%	direct attack	[83]
		*M. aeruginosa* NIES 90				88.8 ± 3.7%		
		*M. aeruginosa* NIES 843				94.6 ± 1.4%		
		*M. aeruginosa* FACHB-905				84.9 ± 0.3%		
		*M. aeruginosa* PCC 7806				86.5 ± 2.1%		
	*S. amritsarensis*	*M. aeruginosa* NIES 44	500 ± 100 *	5%	5 d (10 d)	81.4 ± 0.57% (80.7 ± 0.87%)	NA	[5]
		*M. aeruginosa* NIES 90				51.3 ± 7.83% (80.9 ± 6.49%)		
		*M. aeruginosa* NIES 843				74.6 ± 0.00% (89.8 ± 2.89%)		
		*M. aeruginosa* FACHB-905				85.4 ± 2.21% (98.8 ± 1.05%)		
		*M. aeruginosa* DCM4				83.2 ± 0.00% (96.6 ± 4.79%)		
	*S. jiujiangensis* JXJ 0074	*M. aeruginosa* FACHB-905	5.0 × 10^6^	10%	8 d	90.50 ± 1.08%	indirect attack	[84]
	*Streptomyces* sp. U3	*M. aeruginosa* PCC 1752	NA	5%	3 d	36.22%	indirect attack	[85]
	*Rhodococcus* sp. KWR2	*M. aeruginosa* NIES 843	1.72 × 10^6^	2% (filtrate)	5 d	97%	indirect attack	[34]
		*M. aeruginosa* UTEX 2388				94%		
		*M. aeruginosa* KW				79%		
		*M. aeruginosa* Mi 0601				75%		
	*Microbacterium* sp. F3	*M. aeruginosa* FACHB-905	2.5 × 10^6^	10%	7 d	84.8%	indirect attack	[35]
	*Arthrobacter* sp.	*M. aeruginosa*	2.0 × 10^6^	9%	10 d	32.3 ±13.8%	NA	[14]
**Firmicutes**	*Bacillus subtilis* C1	*M. aeruginosa*	1000 *	1%	2 d	85%	NA	[86]
	*B. fusiformis* B5	*M. aeruginosa*	412.3 *	3.6 × 10^7^ **	7 d	nearly 90%	indirect attack	[87]
	*Bacillus* sp. S51107	*M. aeruginosa* 9110	1.0 × 10^6^	10%	6 d	92.51 ± 2.79%	indirect attack	[88]
		*M. aeruginosa* PCC 7806				91.65 ± 1.00%		
	*Bacillus* sp. AF-1	*M. aeruginosa* NIES 843	1.6 × 10^3^	2%	3 d (6 d)	77% (93%)	indirect attack	[51]
	*Bacillus* sp. Lzh-5	*M. aeruginosa* 9110	1.0 × 10^7^	10%	6 d	91.2 ± 6.3%	indirect attack	[46]
	*Bacillus* sp. T4	*M. aeruginosa* KW	1.0 × 10^6^	5%	3 d	~100%	indirect attack	[48]
	*B. licheniformis* Sp34	*M. aeruginosa* DCM3	1.35 × 10^5^	5%	5 d (10 d)	69.4 ± 0.67 (97.1 ± 0.86%)	indirect attack	[47]
		*M. aeruginosa* DCM4			5 d (10 d)	60.8 ± 1.63 (82.4 ± 2.09)		
		*M. aeruginosa* NIES 843			5 d (10 d)	78.7 ± 5.94% (97.1 ± 0.86%)		
	*B. cereus* DC22	*M. aeruginosa* FACHB-905	1.0 × 10^8^	10%	4 d (7 d)	74.89 ± 2.23% (78.45 ± 0.68%)	NA	[89]
	*Bacillus mycoides* B16	*M. aeruginosa* PCC 7806	~1.0 × 10^6^	NA	6 d	97%	NA	[90]
	*Bacillus methylotrophicus* ZJU	*M. aeruginosa*	1.0 × 10^7^	16.7%	3 d	89 ± 0.5%	indirect attack	[50]
	*Bacillus* sp. Mal 11-2	*M. aeruginosa* NIES 843	NA	6.7%	10 d	up to 60%	NA	[43]
	*Bacillus* sp. Mal 11-10				10 d	55~64%		
	*B. amyloliquefaciens* FZB42	*M. aeruginosa* NIES 843	1.0 × 10^6^	NA	7 d	98.78%	NA	[91]
	*B. amyloliquefaciens* CH03					94.39%	NA	
	*Bacillus* sp. B50	*M. aeruginosa* FACHB-905	NA	10%	5 d	100%	indirect attack	[92,93]
		*M. aeruginosa* FACHB-1023				62.52%		
		*M. aeruginosa* NIES 843				100%		
		*M. aeruginosa* PCC 7806				66.90%		
		*M. aeruginosa* CHAB-439				71.08%		
		*M. aeruginosa* CHAB-456				60.33%		
	*B. amyloliquefaciens* T1	*M. aeruginosa* FACHB-905	1.0 × 10^6^	5%	6 d	99.4%	indirect attack	[49,94]
		*M. aeruginosa* FACHB-907		2%	4 d	76.9 ± 3.1%		[49]
		*M. aeruginosa* FACHB-908		2%	4 d	78.2 ± 2.2%		
		*M. aeruginosa* FACHB-912		2%	4 d	72.9 ± 3.0%		
		*M. aeruginosa* PCC 7806		2%	4 d	85.1 ± 1.8%		
	*B. methylotrophicus* ZJU	*M. aeruginosa*	1.0 × 10^7^	16.7%	3 d	89.0 ± 0.5%	NA	[50]
	*Paenibacillus* sp. SJ-73	*M. aeruginosa* PCC 7806	NA	5%	7 d	83.97 ± 1.60%	indirect attack	[95]
		*M. aeruginosa* TH1701	NA	5% (10%)		92.10% (94.38%)		
	*Exiguobacterium* sp. h10	*M. aeruginosa* PCC 7820	NA	5%	2 d (6 d)	43.4% (73.6%)	indirect attack	[44]
	*Exiguobacterium* sp. A27	*M. aeruginosa* PCC 7806	1.0 × 10^7^	10%	2 d	64.4 ± 10.3%	indirect attack	[96]
		*M. aeruginosa* 9110	NA			58.3 ± 8.2%		
	*Exiguobacterium indicum* EI9	*M. aeruginosa* FACHB-905	4.4 × 10^7^	1.1 × 10^8^ **	NA	NA	NA	[45]
	*Staphylococcus* sp. F1	*M. aeruginosa* FACHB-905	2.5 × 10^6^	10%	7 d	96.0%	indirect attack	[35]
**Thermus**	*Deinococcus metallilatus* MA1002	*M. aeruginosa* PCC 7806	6.0 × 10^6^	10%	3 d	up to 80%	indirect attack	[52]
**Ascomycota**	*Trichoderma citrinoviride*	*M. aeruginosa*	3.2 × 10^4^	10%	2 d	100%	NA	[6]
	*Aspergillus niger* 7806F3	*M. aeruginosa* PCC 7820	5.0 × 10^6^	10%	4 d	up to 80%	indirect attack	[15]
	*Penicillium chrysogenum*	*M. aeruginosa*	NA	3.85%	6 d	69.56%	indirect attack	[97]
	*Aureobasidium pullulans* KKUY070	*M. aeruginosa* DRCK1	5.0 × 10^4^	1.2 × 10^6^ **	1 d (3 d)	84% (100%)	NA	[98]
**Basidiomycetes**	*Lopharia spadicea*	*M. aeruginosa* FACHB-912	798 ± 13 *	NA	39h	100%	NA	[99]
	*Phanerochaete chrysosporium*	*M. aeruginosa*	about 1.57 × 10^7^	500 ***	NA	88.6 ± 0.52%	NA	[100,101]
	*Irpex lacteus* T2b	*M. aeruginosa* PCC 7806	646.25±19.11 *	5%	30h	96.82%	direct attack	[102]
	*Trametes hirsuta* T24		705.19±15.45 *		39h	60.19%		
	*T. versicolor* F21a		701.33±13.50 *		30h	100%		[102,103]
	*Bjerkandera adusta* T1		656.28±26.78 *		39h	98.35%		
	*Phellinus noxius* HN-1	*M. aeruginosa* NIES 843	656.28 ± 26.78 *	NA	NA	NA	NA	[104]
	*Trichaptum abietinum* 1302BG	*M. aeruginosa* FACHB-918	750 *	NA	2 d	100%	direct attack	[105]
		*M. aeruginosa* PCC 7806	1300 *	NA	36h	100%		

NA means the date is not available, not mentioned, or unclear. An asterisk (*) stands for the Chl *a* concentration, μg L^−1^; Two asterisks (**) represent the cell concentrations of anti-cyanobacterial microorganisms, cfu mL^−1^; Three asterisks (***) represent the dry cell weight concentrations of the anti-cyanobacterial microorganisms, mg L^−1^.

**Table 2 microorganisms-10-01136-t002:** Anticyanobacterial substances and their EC_50_ on *M. aeruginosa*.

	Anticyanobacterial Substances	Strain Name	Target Cyanobacterium	Initial Cyanobacterial Cell Density (cells mL^−1^)	EC_50_ (mg L^−1^)	References
**Alkaloids**	Harmane (1-methyl-β-carboline)	*Pseudomonas* sp. K44-1	*M. aeruginosa* NIES 299	NA	NA	[66]
	prodigiosin (C_20_H_25_N_3_O)	*S. marcescen*s LTH-2	*M. aeruginosa* TH1	3.0 × 10^6^	0.048 ± 0.004 (24 h)	[76,109]
			*M. aeruginosa* TH2		0.089 ± 0.011 (24 h)	
			*M. aeruginosa* FACHB-905		0.25 (24 h)/0.16 (72 h)	
		*Hahella* sp. KA22	*M. aeruginosa* FACHB-1752	NA	5.87 (72 h)	[31]
		*S. marcescens* BWL1001	*M. aeruginosa*	NA	NA	[30]
	2-(3, 4-dihydroxy2-methoxyphenyl)-1, 3-benzodioxole-5-carbaldehyde	*Phellinus noxius* HN-1	*M. aeruginosa* NIES 843	656.28 ± 26.78 *	20.6 (72 h)	[104]
	3, 4-dihydroxybenzalacetone(C_10_H_10_O_3_)				5.1 (72 h)	
	Bacilysin (L-alanyl-[2,3-epoxycyclohexanone-4]-L-alanine)	*Bacillus amyloliquefaciens* FZB42	*M. aeruginosa* NIES 843	1.0 × 10^6^	4.13 (96h)	[91]
	tryptamine (C_10_H_12_N_2_)	*Streptomyces eurocidicus* JXJ-0089	NA	NA	3.00 ± 0.09 (72 h)	[38]
	Tryptoline (C_11_H_12_N_2_)				2.54 ± 0.05 (72 h)	
	3-methylindole	*Aeromonas* sp. GLY-2107	*M. aeruginosa* 9110	1.0 × 10^7^	1.10 (24 h)	[69]
	indole-3-carboxaldehyde	*Bacillus* sp. S51107	*M. aeruginosa* 9110	1.0 × 10^6^	6.55 (24 h)	[88]
	2′-deoxyadenosine (C_10_H_13_N_5_O_3_)	*Streptomyces jiujiangensis* JXJ 0074	*M. aeruginosa* FACHB-905	5.0 × 10^6^	6.42 (72 h)	[84]
	adenosine				53.75 (72 h)	
	2, 3-indolinedione	*Shewanella* sp. Lzh-2	*M. aeruginosa* 9110	1.0 × 10^7^	12.5	[27]
	4-hydroxyphenethylamine (C_8_H_11_NO)	*Acinetobacter guillouiae* A2	*M. aeruginosa* FACHB-905	~1.0 × 10^6^	22.5 ± 1.9 (72 h)	[72]
**Fatty acid/Cyclic peptides/peptide derivates**	cyclo(Gly-Pro)	*Stenotrophomonas* sp. F6	*M. aeruginosa* 9110	NA	5.9 (24 h)	[29]
	cyclo(Pro-Phe)	*Bacillus* sp. S51107	*M. aeruginosa* 9110	1.0 × 10^6^	1.85 (24 h)	[88]
	cyclo(4-OH-Pro-Leu) (C_11_H_18_N_2_O_3_)	*Chryseobacterium* sp. GLY-1106	*M. aeruginosa* 9110	1.0 × 10^7^	1.26 (24 h)	[41]
	cyclo(Pro-Leu) (C_11_H_18_N_2_O_2_)				2.70 (24 h)	
	Cyclo(Gly-Pro)	*Bacillus* sp. Lzh-5	*M. aeruginosa* 9110	1.0 × 10^7^	5.7 (24 h)	[46]
	Cyclo(Pro-Val)				19.4 (24 h)	
	cyclo(Gly-Pro)	*Shewanella* sp. Lzh-2	*M. aeruginosa* 9110	1.0 × 10^7^	5.7 (24 h)	[27]
	cyclo(Gly-Phe)	*Aeromonas* sp. GLY-2107	*M. aeruginosa* 9110	1.0 × 10^7^	4.72 (24 h)	[69]
	trans-3-indoleacrylic acid	*Rhodococcus* sp. p52	*M. aeruginosa*	7.3 × 10^6^	NA	[113]
	DL-pipecolic acid				NA	
	L-pyroglutamic acid				NA	
	fusaricidins	*Paenibacillus polymyxa* E681	*M. aeruginosa* KW	2.37 ± 0.15 ×10^7^	NA	[3]
**Protein/Amino acids**	protein	*Raoultella planticola*	*M. aeruginosa* FACHB-905	NA	NA	[70]
		*Aeromonas* sp.				
	L-lysine and L-phenylalanine	*Bacillus amyloliquefaciens* T1	*M. aeruginosa* FACHB-905	1.0 × 10^6^	NA	[94]
	L-valine	*Streptomyces jiujiangensis* JXJ 0074	*M. aeruginosa* FACHB-905	5.0 × 10^6^	NA	[37]
	L-lysine	*Streptomyces phaeofaciens* S-9	*M. aeruginosa* NIES 112	NA	NA	[114]
			*M. aeruginosa* NIES 298			
	lysine	*Aeromonas* sp. FM	*M. aeruginosa* FACHB-905	NA	NA	[115]
**Enzymes**	enzyme	*Streptomyces neyagawaensis*	*M. aeruginosa* NIES 298	NA	NA	[80]
	L-amino acid oxidase	*Aquimarina spongiae*	*M. aeruginosa* MTY01	NA	NA	[77]
	microcystinase A	*Sphingopyxis* sp. C1	*M. aeruginosa* FACHB-905	3.75 × 10^6^	NA	[55]
**Others**	active flocculating substance	*Halobacillus* sp. H9	*M. aeruginosa* PCC 7806	2.0 × 10^7^	NA	[26]
			*M. aeruginosa* TAIHU98			
	clavulanate	*Aeromonas* sp. FM	*M. aeruginosa* FACHB-905	NA	NA	[115]
	biosurfactant	*Bacillus subtilis* C1	*M. aeruginosa*	1000 *	NA	[86]
	lumichrome	*Aeromonas veronii* A134	*M. aeruginosa* MGK	NA	NA	[116]
	triterpenoid saponin (C_42_H_70_O_13)_	*Streptomyces* sp. L74	*M. aeruginosa* FACHB-905	1×10^6^	NA	[33]
	hydroquinone	*Stenotrophomonas* sp. F6	*M. aeruginosa* 9110	NA	0.96 (24 h)	[29]
	nanaomycin A methyl ester	*Streptomyces hebeiensis* YIM 001^T^	*M. aeruginosa* FACHB-905	~1.0 × 10^6^	2.97 (72 h)	[117]

NA means the date is not available, not mentioned or unclear; An asterisk (*) stands for the Chl *a* concentration, μg L^−1^.

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
