# Peer review of "Recent Advances in the Research on the Anticyanobacterial Effects and Biodegradation Mechanisms of Microcystis aeruginosa with Microorganisms"

_microorganisms, 2022, doi:10.3390/microorganisms10061136_

Round 1

Reviewer 1 Report

The reviewer would like to thank the authors for their interesting review on anticyanobacteria and their effects on M. aeruginosa.  The reviewer would like to provide both stylistic and technical notes on the review provided here:

Stylistic:

  1. In general the grammar needs improvement throughout the text.

  1. A small note but it seems on occasion the authors are using the abbreviation sp. for referring to multiple species (plural). There is actually a different, correct plural abbreviation: spp. which should be used in those cases rather than sp.

  1. Word usage at several points is incorrect, for example: lines 98-99

“Obviousness, Bacillus has the potential application for HCBs control”

The correct word choice would be “Obviously”

  1. On a separate, but related topic; terms like “Nowadays” (Line 223) are not the most appropriate for scientific literature. The reviewer would suggest a word like “Currently”, “At present time”, or even “Recent literature suggests” rather than a less formal term like “Nowadays”.

  1. Table 2 contains some Chinese characters in one of the columns. The reviewer is uncertain if the authors would like to keep it or remove it.

  1. The authors may wish to omit the usage of possessive pronouns (our) when referencing their own work. In this review it might not be necessary to do that.

Technical

  1. Table 1 needs to be reworked. The reviewer believes it might be confusing to the reader to see some values listed as cells/mL and others listed as ug/L (for Chl α). The reviewer would suggest adding an asterisk (*) to these values so as not to confuse the reader
  2. Again with Table 2, the dosages have the same issue (switching at times from % to various methods of determining concentrations). The reviewer would suggest again adding an asterisk for Chl α and additionally having a comprehensive caption explaining the various different methods for determining concentration.
  3. The authors may wish to comment on a comparison among various methods of determining concentration and how that relates to potency. The readers may find that helpful.
  4. Table 2 could also use some clarification with concentrations (cells/mL vs ug/L). Please consider revising.
  5. The reviewer wonders if the authors, in section 4.1, could talk about the effectiveness of “indirect attack” versus “direct attack” for anticyanobacteria organisms immobilized in matrices. The reviewer believes that “indirect attack” may work better than “direct attack” in actual applications. Could the authors please comment?

Reviewer 2 Report

This is a comprehensive account of different aspects of inhibition of an important cyanotoxin-producing cyanobacterium M. aeruginosa and elimination of cyanotoxins in water bodies. The detailed summary of the literature available to the date reflects adequately state of the art in this field. Definitely, I would like to see this work published, although it would benefit from some style editing aiming at making the text more condensed.

Round 2

Reviewer 1 Report

This work looks better, there are still a few corrections that could be made:

The Reviewer has just a few more suggestions for grammatical changes:

·       It is hoped to enhance understanding the anticyanobacterial microorganisms and provide  a rational attitude towards the future applications of anticyanobacterial microorganisms.

To

“This review aims to to enhance understanding of anticyanobacterial microorganisms and to provide  a rational approach towards future applications of anticyanobacterial microorganisms.”

·       Hope this review will enhance understanding the anticyanobacterial microorganisms and provide a 61 rational attitude towards future application

to

“This review will enhance understanding the anticyanobacterial microorganisms and provide a rational attitude towards future application”

·       Over the past few decades, isolation and identification of microorganisms that with the anticyanobacterial effects have attracted extensive concern.

to

“Over the past few decades, isolation and identification of microorganisms with anticyanobacterial effects have attracted extensive concern.”

·       More than 50 genera belonging mainly to Proteobacteria, Actinomycetes, Bacteroidetes, Firmicutes and Thermus.

To “This includes more than 50 genera belonging mainly to Proteobacteria, Actinomycetes, Bacteroidetes, Firmicutes and Thermus.”

·       To summarize, the anticyanobacteria can effectively inhibit the growth of M. aeruginosa, and perform the inhibition effect at a low concentration

To “To summarize, the anticyanobacteria can effectively inhibit the growth of M. aeruginosa, and cause an inhibition effect at a low concentration”

·       Compared with the studies of anticyanobacteria, the researches and applications of fungi for eliminating or inhibiting M. aeruginosa cells have not received much attention until 2010

To “Compared with the studies of anticyanobacteria, the research and application of fungi for eliminating or inhibiting M. aeruginosa cells has not received much attention until 2010”

·       The relatively transcriptional level of some critical genes in cyanobacteria can be dramatically changed by anticyanobacterial microorganisms and substances, including genes related to the

To “The relative transcriptional level of some critical genes in cyanobacteria can be dramatically  changed by anticyanobacterial microorganisms and substances, including genes related to the”

The reviewer is unsure about the following phrase:

·       Except for affecting the expressions of above genes, the gene such as mcyB involves in the microcystins synthesis is also inhibited by Penicillium sp. [96], the white-rot fungi P. chrysosporium [99, 100] and P. noxius HN-1 [103];

It might be easier to say,

“Gene such as mcyB that are involved in microcystins synthesis are also inhibited by Penicillium spp. [96], the white-rot fungi P. chrysosporium [99, 100] and P. noxius HN-1 [103];
